# RadarPDR: Radar-Assisted Indoor Pedestrian Dead Reckoning

**DOI:** 10.3390/s23052782

**Published:** 2023-03-03

**Authors:** Jianbiao He, Wei Xiang, Qing Zhang, Bang Wang

**Affiliations:** 1Shenzhen Polytechnic (SZPT), Liuxian Avenue 7098, Shenzhen 518055, China; 2School of Electronic Information and Communications, Huazhong University of Science and Technology (HUST), Luoyu Road 1037, Wuhan 430074, China; 3Hubei International Trade Supply Chain Management Co., Ltd., Wuhan 430000, China

**Keywords:** trajectory correction, radar ranging and correction, pedestrian dead reckoning, indoor tracking

## Abstract

Pedestrian dead reckoning (PDR) is the critical component in indoor pedestrian tracking and navigation services. While most of the recent PDR solutions exploit in-built inertial sensors in smartphones for next step estimation, due to measurement errors and sensing drift, the accuracy of walking direction, step detection, and step length estimation cannot be guaranteed, leading to large accumulative tracking errors. In this paper, we propose a radar-assisted PDR scheme, called RadarPDR, which integrates a frequency-modulation continuous-wave (FMCW) radar to assist the inertial sensors-based PDR. We first establish a segmented wall distance calibration model to deal with the radar ranging noise caused by irregular indoor building layouts and fuse wall distance estimation with acceleration and azimuth signals measured by the inertial sensors of a smartphone. We also propose a hierarchical *particle filter*(PF) together with an extended Kalman filter for position and trajectory adjustment. Experiments have been conducted in practical indoor scenarios. Results demonstrate that the proposed RadarPDR is efficient and stable and outperforms the widely used inertial sensors-based PDR scheme.

## 1. Introduction

Pedestrian dead reckoning (PDR) is a widely-used pedestrian tracking solution in satellite signal-masking environments where global navigation satellite systems such as the global positioning system (GPS) cannot work correctly [1,2,3,4]. The cumulative error problem is the main obstacle for achieving high tracking accuracy in both the Inertial Navigation Systems (INSs) and the Step-and-Heading Systems (SHSs) of PDR [5,6,7]. For example, SHSs adopt the step-and-heading strategy to calculate the step vectors (*step length, heading*) from the inertial sensor data and then evaluate the pedestrian trajectory according to the step vectors in a step-by-step way. However, such recursive estimation accumulates large errors originated from sensing drift with the increase of walking distance. Researchers have devoted a lot of effort towards solving the problem. Their approaches can be generally divided into two categories: (1) mining latent information to correct sensor errors; (2) introducing auxiliary information to adjust deviated trajectory.

Latent information assisted methods utilized prior knowledge or general inference as an extra supplement to fuse inertia step vector estimates for improving pedestrian trajectory accuracy. In foot-mounted INSs, the zero-velocity update (ZUPT) algorithm is a workable solution to limit long-term accumulation of errors [8,9,10,11]. The ZUPT algorithm is based on the fact that the speed of the sole should be zero during the stance phase period, when the sole is completely in contact with the ground. As such, the acceleration integral could be avoided [12]. The building layout is another source of latent information closely related to indoor positioning services, which can affect pedestrian motion characteristics. For instance, pedestrians tend to walk in straight lines in straight corridors [13], or in one of several dominant directions of the buildings [14,15,16]. Pedestrian trajectory and heading direction can be inferred with the help of such layout characteristics and geometries in buildings. However, those inferences are not general enough in many indoor scenarios. For example, a pedestrian could have walked not according to layout dominant directions for a long time [17].

Auxiliary information assisted methods adopt location-specific information inherent in the environment or created by additional technologies to adjust the PDR positioning result via a fusion algorithm. The auxiliary techniques or information include Wi-Fi [18,19,20,21,22,23], Bluetooth [24,25], Ultra-Wideband (UWB) [26,27,28], visible light [17,29], and indoor map  [30,31,32,33,34,35]. Due to the azimuth effect, the procedure of information fusion and trajectory calibration is a nonlinear model that can be solved by particle filtering [20,28,33,34] and extended Kalman filter (EKF) [36,37,38]. The auxiliary information-assisted methods are more applicable in practice, since they do not depend on a specific building layout. However, most of the auxiliaries require additional hardware infrastructure, which often has expensive deployment costs.

With miniaturization, lightweight, and low power consumption of sensors, more sensors can be integrated into mobile terminals and wearable devices. For such considerations, we have initialized experiments of using a lightweight and portable 24 GHz *frequency-modulation continuous-wave radar* (FMCW radar, simplified as radar hereafter) to measure the distance between a radar and a wall. Figure 1 presents our experiment device and experiment scenario, where a student used a stand to connect an external yet small radar with a smartphone. On the one hand, we recognized that such wall distance estimations can be exploited to improve the accuracy of inertial sensor-based PDR. On the other hand, although a radar can estimate the distance to a wall, the estimation could fluctuate severely during pedestrian walking due to the irregular indoor layout, such as doors, windows, corners, obstructions, etc. Therefore, we need to first calibrate wall distance estimations from a radar and next fuse such estimations to adjust step positions.

In this paper, we propose a *Radar-assisted Pedestrian Dead Reckoning* scheme, called RadarPDR, for indoor tracking, which employs an external radar for trajectory adjustment. To the best our knowledge, we are the first to use and experiment with an FMCW radar for indoor pedestrian tracking. In RadarPDR, we first propose a segmented wall distance calibration model, which exploits the space-time correlation of neighbor points to deal with the radar ranging noise caused by irregular indoor building layouts. We next design a hierarchical particle filter together with an extended Kalman filter to fuse calibrated wall distances with acceleration, angular velocity, and azimuth information to adjust pedestrian step positions. We conduct field experiments in typical indoor scenarios. Results validate the effectiveness of our RadarPDR in terms of lower tracking errors compared with the widely used inertial sensor-based PDR scheme. Our main contributions can be summarized as follows:Employ an FMCW ranging radar to assist inertial PDR for indoor pedestrian tracking;Correct distance measurement errors caused by concave/convex wall structure in actual indoor environments;Fuse corrected wall distance, pedestrian step length, and head direction for pedestrian trajectory adjustment.

## 2. Related Works

Radio-frequency (RF) is the most common auxiliary technology, such as Wi-Fi and Bluetooth, to assist PDR indoor tracking. Some research has proposed to integrate fingerprinting-based localization with PDR [18,19,20,21,39]. For example, Shi et al. [19] use a weighted K-Nearest Neighbor algorithm to obtain Wi-Fi RSS (received signal strength) fingerprinting locations, which are fused with PDR locations via a simple weighted average. Some researchers propose to fuse the lidar data and inertial sensor data to solve the accumulative error problem in localization [23,39,40,41]. Liu et al. [39] develop a radio receiver to estimate the Doppler speed and range information from digital terrestrial multimedia broadcasting (DTMB) signals and carry out an extended Kalman filter (EKF) algorithm to fuse the information for the PDR. Li et al. [20] propose to construct a radio map from pedestrian trajectories for RSS fingerprinting, while adjusting trajectory with the aid of Wi-Fi RSS fingerprinting results via selective particle filtering. These methods need to construct a radio map for fingerprnting either by site survey or crowdsourced trajectories.

Some research fuses the trilateral localization method with PDR [22,33,39,40,42,43,44]. Treating iBeacon as an auxiliary, Xia et al. [33] calculate distances between pedestrian and bluetooth anchor nodes based on an RSS path loss model to estimate pedestrian locations by the trilateration method. These locations are fused with PDR results via a particle filter. Tiwari and kumar Jain [22] propose a Heron-bilateration-based position determination (HBPD) technique to locate a mobile device with a Wi-Fi *access point* (AP). They also propose a hybrid indoor localization system (HILS) to integrate the HBPD technique with the PDR method. These methods require enough anchor points with known locations. Xu et al. [34] propose an enhanced particle filter that fuses smartphone built-in sensors’ outputs and the distances in between smartphones. Tao et al. [42] continuously collect light intensity values from smartphones, which are used to detect light source location for PDR correction. De Cock et al. [43] present a smartphone-based multi-floor indoor localization system and Asraf et al. [45] propose a deep-learning PDR framework based on smartphones’ inertial sensors. Tian et al. [28] propose a PF-based fusion algorithm to correct position error due to sensor drift in a PDR-only system. Compared to the path loss model, these methods have higher distance estimation accuracy. However, almost all RF-based auxiliary technologies require pre-deployed hardware infrastructures.

Indoor layout is another piece of frequently used auxiliary information, mainly working in two ways: deterministic trajectory matching [30,31,32] and probabilistic map model [33,34,35]. Lan and Shih [31] correct the drift error of the gyroscope by comparing the geometric similarity between a PDR trajectory and an indoor map. Nguyen-Huu et al. [32] detect a potential landmark based on turning behaviors in corners or door areas and then update the current position with the landmark position for meeting some physical constraints. Zhou et al. [40] use the conditional random field (CRF) model to estimate the optimal landmark sequence passed by pedestrians with the help of a pseudo indoor plan for PDR position correction. On the other hand, probabilistic map models usually occur in PF-based fusion schemes [33,34]. In these methods, an indoor environment is firstly divided into reachable and unreachable areas, which serve as particle weights in the particle filtering process. For example, Koroglu et al. [35] use a continuous probability distribution function to convert a floor plan into a likelihood heat map for weighting particles. However, these methods require a prior knowledge of indoor layouts.

For ranging-assisted indoor PDR, many research studies calculate distances by communicating with surrounding Wi-Fi access points and then integrate the distance measurements with inertial data to improve positioning accuracy [34,46,47,48,49]. For example, Sun et al. [46] perform ranging based on Wi-Fi fine timing measurement technology and further calculate position through their improved trilateral positioning algorithm. The ranging-based positions are eventually fused with inertial PDR results under an EKF framework. Other research attempts to integrate active ranging devices. Sadruddin et al. [50] integrate inertial PDR as an odometry into a body-mounted laser scanner platform for simultaneous localization and mapping, which also enables more accurate pedestrian localization. Mao et al. [51] use a smartphone to measure distances to a surrounding wall. Wu et al. [48] recognize the steady-heading and non-steady-heading human activities to optimize the PDR without any prior knowledge and Zhang et al. [47] use the Wavelet-CNN deep learning network to train the smartphones’ built-in MEMS sensor data for human activity recognition.

This paper utilizes an external FMCW radar to assist PDR tracking. Since such a radar is entirely a self-transmitting and self-receiving device, our RadarPDR scheme can be applied to environments without prior indoor layouts and does not need additional expensive hardware infrastructure.

## 3. Problem Description

We investigate how to use a ranging radar to adjust pedestrian trajectory for indoor tracking. As illustrated in Figure 2, a user holds a smartphone with an FMCW radar, and a sequence of steps represents his trajectory. Let $s_{n}=\left(h_{n},x_{n},y_{n}\right)$ denote the pedestrian pose at the *n*-th step, where $h_{n}$ is the heading orientation and $\left(x_{n},y_{n}\right)$ the step coordinate. The radar is used to measure the distance between the user and a wall. The sampling rate of radar is 10 Hz, which is higher than the pedestrian walking speed. Thus, between two steps, we can have *M* wall distance measurements, denoted by $d_{n,1},\ldots ,d_{n,M}$. Each observation $d_{n,m}$ links an observation point and a wall point. The observation point is the pedestrian pose when this observation occurs. We assume that *M* observation points are evenly distributed within one step. The wall point is a virtual point on the wall surface, and the distance between the two points is $d_{n,m}$.

The proposed RadarPDR adopts a particle filtering framework for trajectory adjustment. Although the principle is straightforward, its implementation faces many practical challenges: From our field experiences, if a user walks strictly in parallel with a flat wall, distance measurements of an FMCW radar have high accuracy, with ranging errors often smaller than 0.2 m. However, such ideal cases are not often in practical indoor environments. On the one hand, a user may not always walk along with a strict line, and some turning steps are not unexpected. Sometimes, other people may pass across the user and a wall, leading to interferences in distance measurements. On the other hand, it is not easy to find a long flat wall in many buildings, as indoor walls often have some concave and convex structures, such as recessing doorways or protruding pillars. Furthermore, when passing by a corner, wall distance measurements have even larger fluctuations, which often leads to poor positional information for failed trajectory adjustment.

Figure 2 plots the distance measurements of a trajectory in our field experiments. On the one hand, it can be seen that irregular wall structures can cause some distance pulses, and wall corners can cause large distance variations. On the other hand, it can be seen that, besides such distance pulses and large variations, some distance measurements can remain relatively stable in a short walking interval. These observations inspire us to design an adaptive scheme to enforce trajectory adjustment: That is, the pedestrian pose $s_{n}$ is subject to adjustment only when the wall distance $d_{n,m}$ contains sufficient positional information. Otherwise, we accept the pose estimation from the underlying dead reckoning system. Now, the research questions can be summarized as follows:When to enforce pedestrian pose adjustment?How to correct measurement for wall distance $d_{n,m}$?How to adjust a pedestrian pose $s_{n}$ based on $d_{n,m}$?

We next present our solution RadarPDR to these questions.

## 4. The Proposed RadarPDR Scheme

### 4.1. Overview

Figure 3 presents the system diagram of our RadarPDR scheme, which consists of four main modules: (i) data sensing and preprocessing; (ii) inertial PDR, (iii) wall model calibration; (iv) multi-source information fusion.

As sources, accelerometer, gyroscope, and radar continuously yield raw observation data according to their acquisition frequency. The data preprocessing module cuts these data streams into step-wise pieces implemented by peak detection [52] based on acceleration observations. Once a new step is completed, these sorts of observations will be distributed into the following processing modules.

Following standard procedures, the inertial PDR module works to estimate step length and heading increment using inertial sensor observations. Step length is estimated by the Weinberg algorithm [53] based on the acceleration series; heading increment is computed based on the azimuth observations from the gyroscope-based orientation sensor embedded in our Android smartphone. The wall model calibration module aims to tackle the corner and barrier problems per our discussion in the previous section. In contrast, the multi-source information fusion module tries to integrate all available information and finally estimates an accurate pedestrian trajectory.

### 4.2. Wall Model Calibration

This module focuses on two realistic challenges in the wall model: the corner problem and the barrier problem. We assign an operation indicator for each step based on corner detection, which works to tackle the adjustment failure in the corner zone by an adaptive operation. Moreover, a wall distance correction algorithm is developed to handle the outlier issue due to walls’ barrier structures.

#### 4.2.1. Operation Indicator Assignment

The proposed scheme is based on a flat-wall observation that the wall is approximately flat during several adjacent steps. A wall parameter tuple (α,β) with slope α the slope and β the intercept can represent the wall model. Long trajectories can be related to segmented flat walls with several corners. For example, Figure 2 depicts a trajectory beside a wall consisting of two flat-segments and one corner. In the non-corner zone, each flat-segment needs independent parameter initialization and an updating procedure. In addition, wall distances obtained in the corner zone show significant fluctuation. In short, a corner and non-corner zone need different operations in the following modules. Therefore, we generate an operation indicator for every step based on corner detection. In RadarPDR, each step belongs to one of three states: operate, suspend, and reset. The operation of each state is as follows: (i) trajectory adjustment and wall parameter updating; (ii) dead reckoning; (iii) dead reckoning and wall parameter initialization.

The indicator assignment principle is as follows: Firstly, if the heading increment $\theta_{n}$ is more significant than an angle threshold $T_{\theta }$, then *n*-step is judged to be in a corner zone, and its corresponding indicator $i_{n}$ is assigned to *suspend*; otherwise we think that *n*-step is in a flat-segment zone, and it needs further judgment. Then, if *n*-step belongs to the beginning $N_{i}$ steps of the flat-segment, $i_{n}$ is assigned to *reset*; otherwise, $i_{n}$ is set to *operate*. Figure 4 presents wall distance observations, heading orientations, and corresponding indicators of several steps around a corner, where $N_{i}=1$ and $T_{\theta }$ = $25^{\circ }$. The $T_{\theta }$ is chosen according to our experiences and has been proven practicable in our experiments to detect whether a pedestrian has cornered.

#### 4.2.2. Wall Distance Correction

In this paper, barrier refers to the convex and concave structures embedded in a flat wall, such as recessing doorways or protruding pillars. These unexpected barriers violate the flat-wall model, yielding noises in wall observations. Figure 5 depicts a series of noisy wall distance data measured in a field experiment. We classify these noises into two categories: (i) random noise; (ii) impulse noise. The random noise comes from inherent radar ranging error, just like many other sensors. It has a relatively small amplitude; the impulse noises come from internal signal fluctuation or wall barrier structures, presenting a much larger amplitude than random noise. Our distance correction algorithm discussed in this subsection aims to handle both random and impulse noises to ensure that the flat-wall model is well satisfied.

Firstly, we discuss the random noise, which can be approximated to Gaussian white noise. To simplify the description, we assume the number of single-step observations M=1 and simplify the notation of the single observation $d_{n,1}$ at the *n*-th step to $d_{n}$. The general situation (M>1) will be explained later. Let $\nu_{n}$ and $q_{n}$ denote the change rate of $d_{n}$ and the process noise. Let $dist_{n}$ and $r_{n}$ denote the wall distance observation and the random noise following a Gaussian distribution with zero mean and $R_{n}$ variance, respectively. The random noise can be corrected by a Kalman filter (KF) with the following state-space model:

Prediction:(1)$$\begin{array}{c} d_{n}=d_{n-1}+\nu_{n} \end{array}$$(2)$$\begin{array}{c} \nu_{n}=\nu_{n-1}+q_{n} \end{array}$$

Observation:(3)$$\begin{array}{c} dist_{n}=d_{n}+r_{n} \end{array}$$

The prediction functions describe the state-transition procedure of wall distance and its change rate, and they are designed by the fact that pedestrians tend to walk in a straight line or change their directions slowly in most cases. When a pedestrian walks in a straight line, the wall distance change rate $\nu_{n}$ remains unchanged, while the process noise $q_{n}∼N\left(0,Q_{n}\right)$ gives a fuzzy range for the direction changing case.

Considering the impulse noise denoted by $I_{n}$, the above observation function is rewritten as
(4)$$dist_{n}=d_{n}+r_{n}+I_{n}$$

Due to the lack of precise statistics of $I_{n}$, the impulse noise cannot be eliminated easily by general Kalman filtering as the random noise is. Thus, we design a two-stage correction strategy. That is, for each observation $dist_{n}$, we judge whether it is an impulse noise firstly, and then select the correction method correspondingly.

The judgment stage is based on a threshold method imposed on the wall distance change rate of $\nu_{n}$. On the one hand, a normal walking trajectory can be regarded as a continuous curve with a relatively large curvature radius, usually a few meters, except in the corner zone, which leads to relatively small $\nu_{n}$ in inlier observations. On the other hand, a much larger amplitude of impulse noise or outlier observation makes the wall distance change rate estimation ${\hat{\nu }}_{n|n}$ severely deviate its normal range. In short, there is a huge margin between the change rate of inlier wall distance and that of the outlier one, as shown in Figure 6. Therefore, we define the outlier detection criterion as
(5)$${\hat{\nu }}_{n|n}\geq T_{\nu }$$
where $T_{\nu }$ is the upper limit of the change rate of the inlier wall distance selected in the margin.

The correction stage adopts the KF solution with the state-space model given by Equations (1)–(3), if there is no outlier. Otherwise, it will switch to a modified KF version. In our problem, a KF’s observation is much less reliable than its prediction when an outlier occurs. The prediction is close to the ground truth in a short time due to the slow-change feature of wall distance change rate. Thus, we drop the noisy observation directly and use modified prediction functions to obtain wall distance correction. The modification lies in the fact that the trend instead of the single estimation of $\nu_{n}$ is used here. The final correction is summarized as
(6)$$\begin{array}{c} {\bar{d}}_{n}=\left\{\begin{array}{cc} {\hat{d}}_{n|n}, & \mathrm{if}\;{\hat{\nu }}_{n|n}<T_{\nu }\;(\mathrm{inlier}\;\mathrm{case}),\\ {\hat{d}}_{n-1|n-1}+{\tilde{\nu }}_{n}, & \mathrm{if}\;{\hat{\nu }}_{n|n}\geq T_{\nu }\;(\mathrm{outlier}\;\mathrm{case}), \end{array}\right. \end{array}$$
where ${\tilde{\nu }}_{n}$ denotes the trend of a few recent distances at the *n*-th step, calculated by an exponential average of wall distance change rate corrections.

The proposed wall distance correction algorithm in a general case is summarized in Algorithm 1. Note that the judgment-correction procedure is just simply repeated *M* times according to the order of occurring, when we have *M* observations in one step.
**Algorithm 1** Wall distance correction algorithm**Input:** {distn,m}n=1:N,m=1:M: wall distance observations**Output:** {d¯n,m}n=1:N,m=1:M: wall distance corrections**for** each step n∈1,2,…N **do**   **for** each observation m∈1,2,…M **do**       1. Get state estimations by standard Kalman filtering:           d^n,m|m, ν^n,m|m = KF (distn,m)       2. Detect outlier and compute correction:       **if** ν^n,m|m<Tν **then**           d¯n,m=d^n,m|m, ν¯n,m=ν^n,m|m       **else**           d¯n,m=d^n,m−1|m−1+ν˜n,m, ν¯n,m=ν˜n,m       **end if**       3. Update trend:           ν˜n,m+1=γν˜n,m+(1−γ)ν¯n,m   **end for****end for**

### 4.3. Multi-Source Information Fusion

After processing by the upper modules, for each step, we now have information including step length $l_{n}$, heading increment $\theta_{n}$ and wall distance $d_{n,m}$. We next fuse them to estimate pedestrian pose $s_{n}$. Based on the Bayes filtering framework, we define a motion model and an observation model to output their relation. The motion model presents the recurrence relation of pedestrian pose $s_{n}$ with step length $l_{n}$ and heading increment $\theta_{n}$ as follows:(7)$$\begin{array}{c} s_{n}=f\left(s_{n-1},l_{n},\theta_{n}\right)=\left[\begin{array}{c} h_{n-1}+\theta_{n}\\ x_{n-1}+l_{n}\sin h_{n}\\ y_{n-1}+l_{n}\cos h_{n} \end{array}\right]. \end{array}$$

As shown in Figure 7, given the pedestrian pose $s_{n}$, the ranging direction $r_{n}$ and the wall distance $d_{n}$, the wall point $\left(x_{n}^{w},y_{n}^{w}\right)$ is represented as
(8)$$\begin{array}{c} x_{n}^{w}=x_{n}+d_{n}\sin r_{n},\;y_{n}^{w}=y_{n}+d_{n}\cos r_{n}. \end{array}$$

Combining the wall model $y=\alpha_{n}x+\beta_{n}$ and the wall point $\left(x_{n}^{w},y_{n}^{w}\right)$, $d_{n}$ is represented as
(9)$$\begin{array}{c} y_{n}+d_{n}\cos r_{n}=\alpha_{n}\left(x_{n}+d_{n}\sin r_{n}\right)+\beta_{n}, \end{array}$$
(10)$$\begin{array}{c} d_{n}=\frac{\alpha_{n}x_{n}+\beta_{n}-y_{n}}{\cos r_{n}-\alpha_{n}\sin r_{n}}. \end{array}$$

Note that the ranging direction is always perpendicular to the pedestrian heading in our device setup. To replace $r_{n}$ with the heading direction $h_{n}$ under the relationship $h_{n}=r_{n}-\frac{\pi }{2}$, $d_{n}$ is then represented as
(11)$$\begin{array}{c} d_{n}=\frac{\alpha_{n}x_{n}+\beta_{n}-y_{n}}{\cos\left(h_{n}+\frac{\pi }{2}\right)-\alpha_{n}\sin\left(h_{n}+\frac{\pi }{2}\right)}=\frac{y_{n}-\alpha_{n}x_{n}-\beta_{n}}{\sin h_{n}+\alpha_{n}\cos h_{n}}. \end{array}$$

In short, the observation model gives wall distance $d_{n,m}$ at observation point $s_{n,m}$ with wall parameters $o_{n}$ as follows:(12)$$\begin{array}{c} d_{n,m}=g\left(s_{n,m},o_{n}\right)=\frac{y_{n,m}-\alpha_{n}x_{n,m}-\beta_{n}}{\sin h_{n}+\alpha_{n}\cos h_{n}}. \end{array}$$

Now, it can be summarized that $l_{n}$, $\theta_{n}$, $d_{n,m}$ are the given information inherited from the upper modules, while $s_{n}$ or $s_{n,m}$, $o_{n}$ are the unknown states to be estimated in this module (note that $s_{n}=s_{n,1}$). Furthermore, in the observation model, if $s_{n,m}$ serves as the state to be estimated, then $o_{n}$ needs to be the known parameter and vice versa. Therefore, we adopt a PF-EKF architecture to solve this dual-estimation problem, where PF takes charge of trajectory adjustment for $s_{n}$, while EKF is in charge of wall parameter estimation for $o_{n}$.

#### 4.3.1. Trajectory Adjustment

The basic idea of PF-based trajectory adjustment [54] is as follows: During each round of iteration, PF makes a group of guesses, known as particles, around the unreliable state estimated by the motion model. This is the particle sampling procedure. Some of the particles are positive adjustments, closer to the ground truth, while others are negative since they have large deviations. We propose a judgment procedure to weight the particles. Finally, another sampling procedure is implemented on the weighted particles. Those with high weights are more likely to be selected, leading to an improvement in pedestrians’ pose estimation. We next explain these three steps in details below.

The sampling procedure is performed on step length $l_{n}$ and heading increment $\theta_{n}$ from which the errors of the pose originate. We assume that the errors of $l_{n}$ and $\theta_{n}$ follow Gaussian distribution with zero mean and $\sigma_{l},\sigma_{\theta }$ standard deviation, respectively:(13)$$\begin{array}{c} \delta l_{n}^{\left[k\right]}∼N\left(0,\sigma_{l}^{2}\right),\;\delta \theta_{n}^{\left[k\right]}∼N\left(0,\sigma_{\theta }^{2}\right) \end{array}$$

Then, the corrections are represented by
(14)$$\begin{array}{c} l_{n}^{\left[k\right]}=l_{n}-\delta l_{n}^{\left[k\right]},\;\theta_{n}^{\left[k\right]}=\theta_{n}-\delta \theta_{n}^{\left[k\right]} \end{array}$$
where *n* is the step index and *k* is the particle index. According to the motion model, particle pose is reckoned as
(15)$$\begin{array}{c} s_{n}^{\left[k\right]}={\left[\begin{array}{c} h_{n}^{\left[k\right]},x_{n}^{\left[k\right]},y_{n}^{\left[k\right]} \end{array}\right]}^{T}=f\left(s_{n-1},l_{n}^{\left[k\right]},\theta_{n}^{\left[k\right]}\right). \end{array}$$

*K* particles are generated within one step by repeating the above procedure.

The essence of weight calculation is to judge the reliability of particles. Each particle can obtain its wall distance prediction ${\hat{d}}_{n,m}^{\left[k\right]}$ by feeding its observation point $s_{n,m}^{\left[k\right]}$ and wall parameters to the observation model. At the same time, a distance observation $d_{n,m}$ obtained by radar is available. By calculating the proximity between ${\hat{d}}_{n,m}^{\left[k\right]}$ and $d_{n,m}$, we can obtain the particle weight $w_{n,m}^{\left[k\right]}$. This procedure is formulated as
(16)$$\begin{array}{c} {\hat{d}}_{n,m}^{\left[k\right]}=g\left(s_{n,m}^{\left[k\right]},o_{n}\right) \end{array}$$
(17)$$\begin{array}{c} w_{n,m}^{\left[k\right]}=\frac{1}{\sqrt{2\pi S_{n}}}\exp\frac{-{\left(d_{n,m}-{\hat{d}}_{n,m}^{\left[k\right]}\right)}^{2}}{2S_{n}} \end{array}$$where $S_{n}$ is the variance of distance observation residual $|d_{n,m}-{\hat{d}}_{n,m}^{\left[k\right]}|$ at the *n*-step and $o_{n}$ is the wall parameter, and both of them are calculated in the EKF part. The weight $w_{n,m}^{\left[k\right]}$ considers the distance observation residual following a Gaussian distribution with zero mean and $S_{n}$ variance, a larger residual meaning a lower weight. When we have *M* observations in the *n*-th step, we will obtain *M* weights for one particle. Then, the weight of the *k*-th particle at the *n*-th step is calculated by an average of these *M* weights:(18)$$\begin{array}{c} w_{n}^{\left[k\right]}=\frac{1}{M}\sum_{m=1}^{M}w_{n,m}^{\left[k\right]} \end{array}$$

Resampling is a two-sided procedure which aims to ensure that higher weights get resampled more while maintaining particle diversity. We adopt the systematic resampling algorithm introduced in [55]. After the resampling procedure, the final estimated pose state is obtained by averaging the resampled particles.

#### 4.3.2. Wall Parameters Estimation

Due to the absence of a prior map information, the wall parameters necessary for particle weighting are unknown to us initially. The proposed wall parameter estimation algorithm consists of two phases: the initialization phase and the updating phase. The initialization phase happens at the beginning $N_{i}$ steps of each flat-segment, while the updating phase happens at the remaining steps of that flat-segment.

We first introduce the initialization phase. Although the EKF algorithm is insensitive to initial values, wall parameters from random initialization may deviate from the ground truth a lot, which could distort the PF weighting procedure. To deal with such issues, we design an initialization strategy based on the flat-wall discovery as follows: Given an observation point $\left(h_{n},x_{n,m},y_{n,m}\right)$ and wall distance observation $d_{n,m}$, its corresponding wall point $\left(x_{n,m}^{w},y_{n,m}^{w}\right)$ can be calculated by the following geometric relationship:(19)$$\begin{array}{c} x_{n,m}^{w}=x_{n,m}+d_{n,m}\cos\left(h_{n}\right) \end{array}$$(20)$$\begin{array}{c} y_{n,m}^{w}=y_{n,m}-d_{n,m}\sin\left(h_{n}\right). \end{array}$$

The wall parameters are estimated by using the least-square method with all wall points within this phase. In this phase, pedestrian poses and observation points are estimated by dead reckoning via the motion model, since the PF-EKF architecture does not work without wall parameters.

We realize a step-by-step online estimation for wall parameters via extended Kalman filtering with pedestrian pose and wall distance observations during the updating phase. Let $o_{n}=\left[\alpha_{n},\beta_{n}\right]$ denote the wall parameters at the *n*-th step. According to the flat-wall model, $o_{n}$ changes little in several neighboring steps. We thus have the following state transition function:(21)$$\begin{array}{c} o_{n}=F_{n}o_{n-1}+w_{n}, \end{array}$$
where $F_{n}$ is a two-dimensional identity matrix. The observation function is exactly the observation model given by Equation (Equation 12), rewritten as
(22)$$\begin{array}{c} d_{n}=g\left(s_{n};o_{n}\right)+v_{n}. \end{array}$$

Note that here $s_{n}$ is considered as a known parameter from the PF. $w_{n}$ and $v_{n}$ are independent Gaussian noise processes with zero mean and $W_{n}$ and $V_{n}$ covariance matrices, respectively.

The extended KF utilizes only the estimated state ${\hat{o}}_{n-1|n-1}$ of the previous step and the current observation $d_{n}$ to estimate the current state ${\hat{o}}_{n|n}$, and the update procedures at the *n*-th step are formulated below:(23)$$\begin{array}{c} {\hat{o}}_{n|n-1}=F_{n}{\hat{o}}_{n-1|n-1} \end{array}$$(24)$$\begin{array}{c} P_{n|n-1}=F_{n}P_{n-1|n-1}F_{n}^{T}+W_{n} \end{array}$$(25)$$\begin{array}{c} S_{n}=G_{n}P_{n|n-1}G_{n}^{T}+V_{n} \end{array}$$(26)$$\begin{array}{c} K_{n}=P_{n|n-1}G_{n}^{T}S_{n}^{-1} \end{array}$$(27)$$\begin{array}{c} {\hat{o}}_{n|n}={\hat{x}}_{n|n-1}+K_{n}\left(d_{n}-g\left(s_{n};{\hat{o}}_{n|n-1}\right)\right) \end{array}$$(28)$$\begin{array}{c} P_{n|n}=\left(I-K_{n}G_{n}\right)P_{n|n-1} \end{array}$$
where $S_{n}$ is the distance observation residual. $G_{n}$ is the observation matrix that is equal to the Jacobi matrix of $g\left(s_{n};o_{n}\right){|}_{o_{n}={\hat{o}}_{n|n-1}}$,
(29)$$\begin{array}{c} G_{n}=\left[\begin{array}{cc} \frac{\left(\beta_{n}-y_{n}\right)\cos h_{n}-x_{n}\sin h_{n}}{D^{2}} & \frac{1}{D} \end{array}\right]{|}_{o_{n}={\hat{o}}_{n|n-1}} \end{array}$$
where $D=\sin h_{n}+\alpha_{n}\cos h_{n}$. *I* is an identity matrix. Just like the processing style in wall distance correction, we repeat the update procedure *M* times according to the occurring order of these *M* wall distance observations at one step.

In the PF part, wall parameters must ensure pedestrian poses, while this is the exact opposite in the EKF part. Consequently, these two parts are concatenated to boost each other, as illustrated in Figure 8. At the iteration of *n*-step, the step length, heading increment, wall distance observations from previous modules and wall parameters, and residual covariance from the EKF are fed to the PF, which outputs pedestrian pose estimation after the three procedures of new particles’ sampling, weighting, and resampling. Then, the pose estimated by PF and wall distance observations are fed into the EKF, which outputs updated wall parameters. The complete trajectory adjustment algorithm is summarized in Algorithm 2.
**Algorithm 2** Trajectory adjustment algorithm**Input:** ${\left\{l_{n}\right\}}_{n=1}^{N}$: step lengths,  ${\left\{\theta_{n}\right\}}_{n=1}^{N}$: heading increments,  ${\left\{d_{n,m}\right\}}_{n=1:N,m=1:M}$: wall distance,
  ${\left\{i_{n}\right\}}_{n=1}^{N}$: indicators
**Output:** ${\left\{s_{n}=\left(h_{n},x_{n},y_{n}\right)\right\}}_{n=1:N}$: pose estimations
1: **for** each step n∈1,2,…N **do**
2:     **if** the indicator $i_{n}$ is *operate* **then**
3:         **for** each particle k∈1,2…,K **do**
4:              Get its pose $s_{n}^{\left[k\right]}$ according to Equations (Equation 13)–(Equation 15)
5:              Get its weight $w_{n}^{\left[k\right]}$ according to Equations (Equation 16)–(Equation 18)
6:         **end for**
7:         Perform Resampling
8:         Get pose estimation $s_{n}$ by average of resampled particles
9:         Update wall parameters $o_{n}$ according to Equations (Equation 21)–(Equation 29)
10:     **end if**
11:     **if** the indicator $i_{n}$ is *suspend* **then**
12:         Get pose estimation $s_{n}$ by dead reckoning motion model
13:     **end if**
14:     **if** the indicator $i_{n}$ is *reset* **then**
15:         Get pose estimation $s_{n}$ by dead reckoning motion model
16:         Initialize wall parameters
17:     **end if**
18: **end for**


## 5. Experiments

### 5.1. Experiment Setup

We used Distance2Go (D2G), an FMCW radar module by Infineon, to acquire wall distances. The radar is powered by a highly integrated transceiver BGT24MTR11, operating at 24–24.2 GHz. The module is connected to a smartphone via a USB cable, ensuring data exchange and power supply. We developed our data acquisition app running on a smartphone to ensure that the radar module and the other two embedded sensors are operated in a synchronized way when collecting measurements. The sampling rate for wall distance is 10 Hz. The inertial data are obtained from an accelerometer and gyroscope embedded in our Honor 8 smartphone with a sampling rate at 20 Hz.

As shown in Figure 1, we adopted a supporting stand to fix the D2G module on a smartphone, keeping a constant 90${}^{\circ }$ angle between heading and ranging direction. The experiments were conducted on the fifth floor of a classroom building at our university. We designed two different routes, as shown in Figure 9, to experiment our scheme. The L-shape and Z-shape route are combinations of straight lines and right angles, and their lengths are 192 m and 136.8 m, respectively. A student holding the device repeats each route 20 times.

To verify the effectiveness of RadarPDR scheme and its modules, we design the following comparison schemes:PurePDR: The traditional inertial pedestrian dead-reckoning, using the motion model to estimate pedestrian trajectory;RadarPDR w/o DC: The RadarPDR scheme without the wall distance correction (DC) module;RadarPDR w/o PU: The RadarPDR scheme without the wall parameter updating (PU) module.

### 5.2. Wall Distance Results

In order to test the effectiveness of the wall distance correction module, we apply two different algorithms to correct wall distance observations on 20 trajectories of each route, including DC in RadarPDR and KF in RadarPDR w/o DC. Specifically, DC is the completed method summarized in Algorithm 1 while KF is the standard Kalman filter given by Equations (Equation 1)–(Equation 3) without the proposed outlier correction stage. Their parameters are set as: process noise variance $Q_{n}=0.01^{2}$, observation noise variance $R_{n}=0.2^{2}$, wall distance change rate limit $T_{\nu }=0.01$, and smoothing factor γ=0.9. For KF, the process noise variance $Q_{n}$ is the same as DC, and the observation noise variance $R_{n}=2^{2}$, which compensates for its weakness in handling outliers.

Since the PDR tracking is a recursive process, a small number of wall distance outliers can have a big impact on the whole trajectory position estimates. Therefore, the error range is more representative of an error correction algorithm’s contribution to the whole system. Figure 10 presents wall distance error range by different algorithms. Although the observation noise setting of KF is much larger than DC, its correction effect is still limited, and the noise range by our scheme has been reduced by 30.7% and 40.7%, in the L-shape and Z-shape route, respectively. DC’s correction performance is even better, reducing the noise by 93.2% and 96.6% in the two routes.

### 5.3. Wall Parameters’ Results

To demonstrate the effectiveness of the wall parameter estimation module, we compare the parameter estimation results of the RadarPDR and RadarPDR w/o PU scheme. The RadarPDR has an initialization phase and an updating phase, whose parameters are set as: the number of initialization steps $N_{i}=4$, the process noise of wall estimation $W_{n}={\left[\left[0.01,0\right],\left[0,0.1\right]\right]}^{T}$, and the observation noise $V_{n}=2.4$. The RadarPDR w/o PU has only the initialization phase with the same parameter setting as RadarPDR. Since the range of wall parameter variables is infinite, the estimation error is not workable as an evaluation criterion here. We define the *wall distance prediction error* (WDPE) to evaluate the estimation performance by
(30)$$WDPE=\sum_{m=1}^{M}g\left(s_{n,m}^{g},o_{n}\right)-d_{n,m}^{g}$$
where $o_{n}$ is the wall parameter estimation at the *n*-th step; $s_{n,m}^{g}$, $d_{n,m}^{g}$ are the ground truth of the *m*-th observation point and wall distance at the *n*-th step.

Figure 11 shows the variation trend of the WDPE as the pedestrian moves. According to the observation model, the wall parameter estimation performance depends on wall distance observation and pedestrian pose estimation. After the wall distance correction, its noise level on the entire trajectory is almost the same, while the pedestrian pose estimations have a cumulative error effect. Therefore, the variation trend of WDPE along with the trajectory mainly depends on the cumulative error effect.

Firstly, we analyze the distribution characteristics of the initial WDPE of each flat-segment in a trajectory. The wall parameters from the initialization phase calculate the initial WDPEs. Thus, they are almost the same in the RadarPDR and RadarPDR w/o PU scheme. The initial WDPE in the first flat-segment is relatively small, while quite larger in the next flat-segments. This is because the pedestrian pose errors are small at the beginning of the trajectory and gradually increase as the walking distance increases, resulting in a large wall parameter re-initialization error after the corner zone. Then, we compare the WDPE of different schemes in each segment. In the first flat-segment of the L-shape and Z-shape route, the amplitude of WDPE in RadarPDR is slightly larger than that in RadarPDR w/o PU, which can be attributed to the cumulative error effect of pedestrian pose used in EKF updating. In the next flat-segments, however, the WDPE converges to a small amplitude in the RadarPDR but can only change linearly with a flawed initial direction in the RadarPDR w/o PU. This is because RadarPDR considers more observations for wall parameter updating, which is able to reduce the error in the initialization phase as only a few initial observations are used. Furthermore, the initial direction in the RadarPDR w/o PU can easily deviate from ground truth according to Figure 11.

### 5.4. Trajectory Adjustment Results

In this experiment, we compare the performance of four different trajectory estimation/correction schemes. The PurePDR scheme only uses the motion model for trajectory estimation. Moreover, the rest are the proposed RadarPDR and its variants. They have the same parameters setting in the PF part: the number of particles K=128, the number of initialization steps $N_{i}=4$, the error scale of step length $\sigma_{l}=0.4$ and heading increment $\sigma_{\theta }=0.2$.

Figure 12 plots the adjusted trajectory results and their corresponding CDFs for L-shape and Z-shape routes. Compared with the PurePDR scheme, the RadarPDR w/o DC produces a comparable average localization error (ALE) in the L-shape route and even performs much worse in the Z-shape trajectory, which can be attributed to the large wall distance error deactivating the trajectory adjustment procedure. This observation also verifies the importance of the DC module. On the other hand, the RadarPDR w/o PU can achieve smaller ALE on both the L-shape and Z-shape routes. Moreover, the complete RadarPDR has the best trajectory adjustment performance, reducing its ALE by 48.1% in the L-shape route and 41.1% in the Z-shape route. It is also worth mentioning that the PurePDR overtakes the RadarPDR at the point (13.5 m, 0.98) in Figure 12d, resulting from one failed trajectory adjustment in the 20 tests on the Z-shape route. Combined with the analysis of WDPE in the previous section, this failure should be attributed to large initial wall parameter error, which makes the EKF estimation process divergent and leads to the wrong trajectory correction direction.

## 6. Conclusions

In this paper, we have reported our initial attempts of applying an external FMCW radar to assist PDR-based indoor pedestrian tracking. Although such a ranging radar can measure wall distances, its measurements during walking could come with large errors due to complicated indoor layouts and diverse trajectory types. The proposed RadarPDR deploys two algorithms addressing wall distance correction and trajectory adjustment. The trajectory adjustment algorithm fuses step lengths and heading increments from accelerometers and gyroscopes with wall distance from the radar, where the EKF takes charge of wall parameter estimation based on wall distance observations and pedestrian locations. Simultaneously, the PF utilizes outputs of the EKF and distance observations to adjust pedestrian trajectory. Our field experiments have validated the effectiveness of the proposed RadarPDR in terms of higher tracking accuracy. 

## Figures and Tables

**Figure 1 sensors-23-02782-f001:**
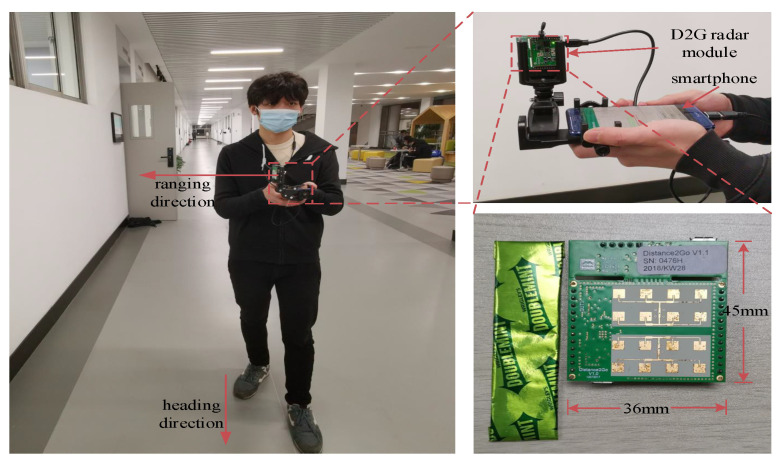
The FMCW radar and experiment scenario.

**Figure 2 sensors-23-02782-f002:**
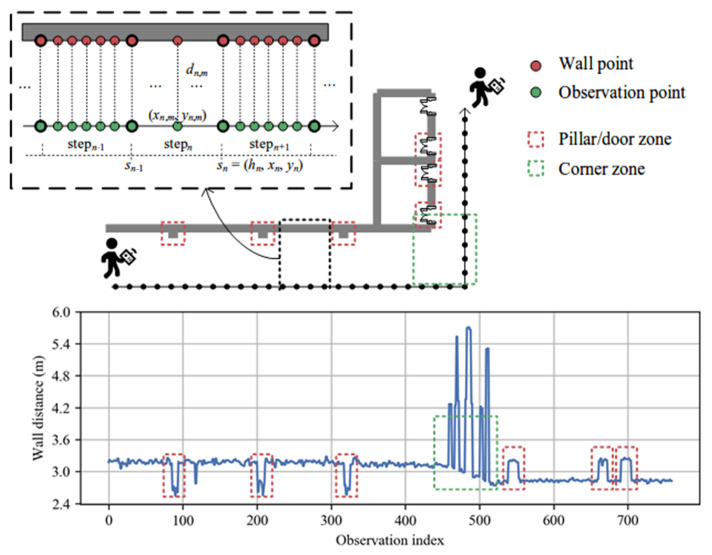
Problem description. The user trajectory is represented by a sequence of pedestrian poses. Each pose $s_{n}$ is denoted as a tuple $\left(h_{n},x_{n},y_{n}\right)$ with $h_{n}$ the heading orientation and $\left(x_{n},y_{n}\right)$ the step coordinate. The wall distance between adjacent steps establishes the positional relationship between pedestrian pose and the surrounding walls. Each wall distance $d_{n,m}$ links an observation point on the trajectory and a wall point on the wall surface. The wall distance observations suffer from large fluctuations due to corner and barrier (e.g., pillar or door) zones on the wall.

**Figure 3 sensors-23-02782-f003:**
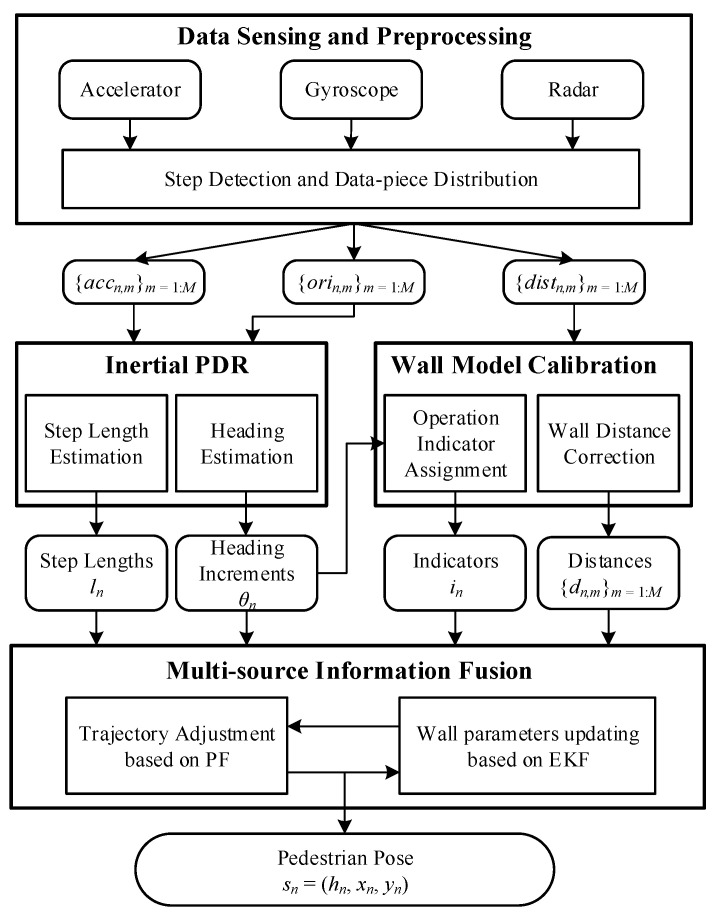
System diagram.

**Figure 4 sensors-23-02782-f004:**
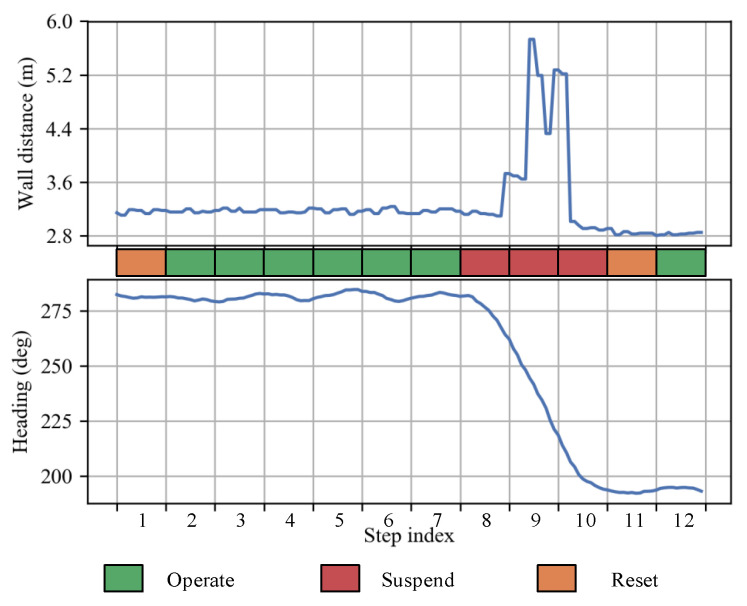
Illustration of the operation indicator assignment strategy. The plots show wall distance observations (above) and heading orientations (below) of several steps around a corner in a typical trajectory. *Suspend* is assigned to corner steps where heading increment $\theta_{n}>$ threshold $T_{\theta }$. *Reset* is assigned to the beginning $N_{i}$ steps of each non-corner segment. Other steps are indicated as *Operate* state.

**Figure 5 sensors-23-02782-f005:**
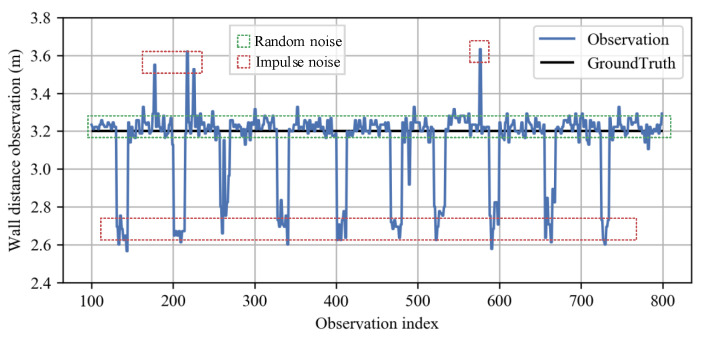
Classification of wall distance observation noise. The plot shows wall distance observations in a straight walking test along a parallel wall. The random noise with small amplitude comes from inherent radar ranging error, while the impulse noise with larger amplitude comes from signal fluctuation or wall barrier structures.

**Figure 6 sensors-23-02782-f006:**
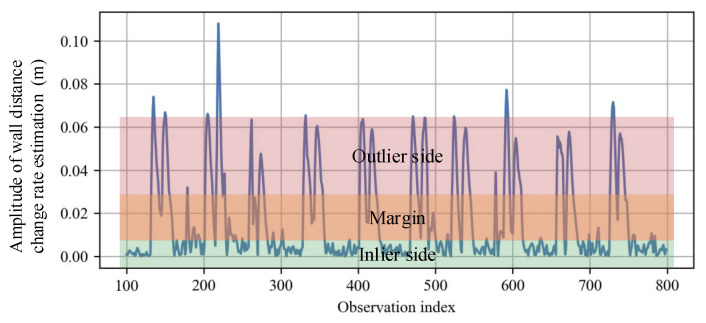
Judgment margin. The plot shows the amplitude of wall distance change rate ${\hat{\nu }}_{n|n}$ estimated on the wall distance observations in Figure 5. The amplitudes of outlier observations are much greater than the inlier ones, thus the judgment threshold $T_{\nu }$ is easily selected from the margin area.

**Figure 7 sensors-23-02782-f007:**
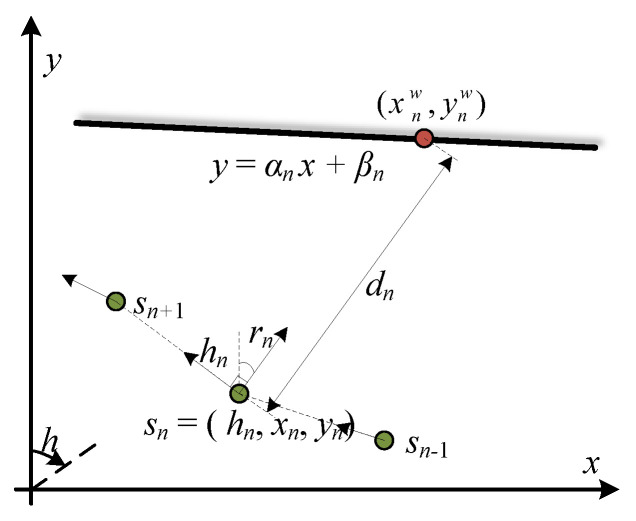
Illustration of the observation model. The wall distance $d_{n}$ is the straight-line distance from the observation point $s_{n}=\left(h_{n},x_{n},y_{n}\right)$ to the wall point $\left(x_{n}^{w},y_{n}^{w}\right)$ along ranging direction $r_{n}$.

**Figure 8 sensors-23-02782-f008:**
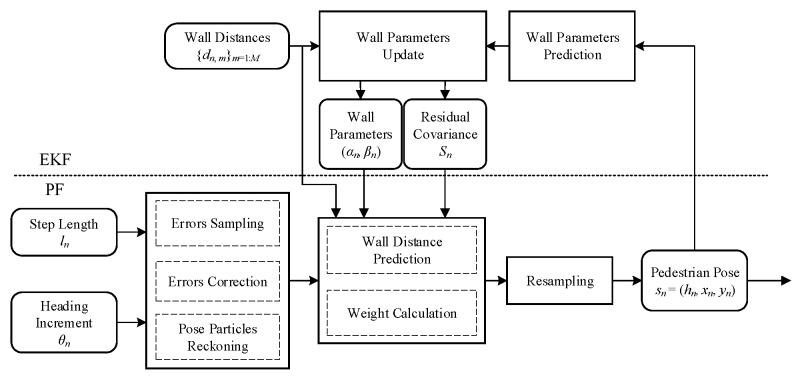
Flow chart of the PF-EKF algorithm.

**Figure 9 sensors-23-02782-f009:**
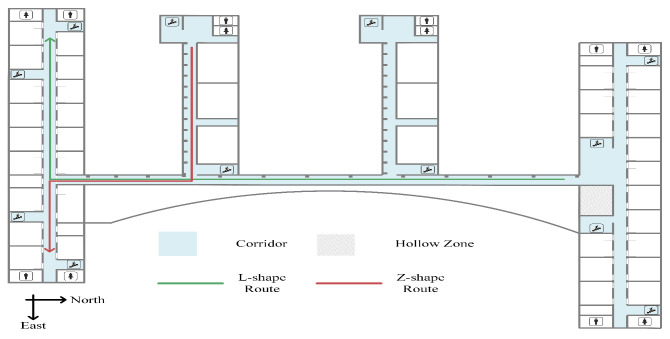
Experiment environment.

**Figure 10 sensors-23-02782-f010:**
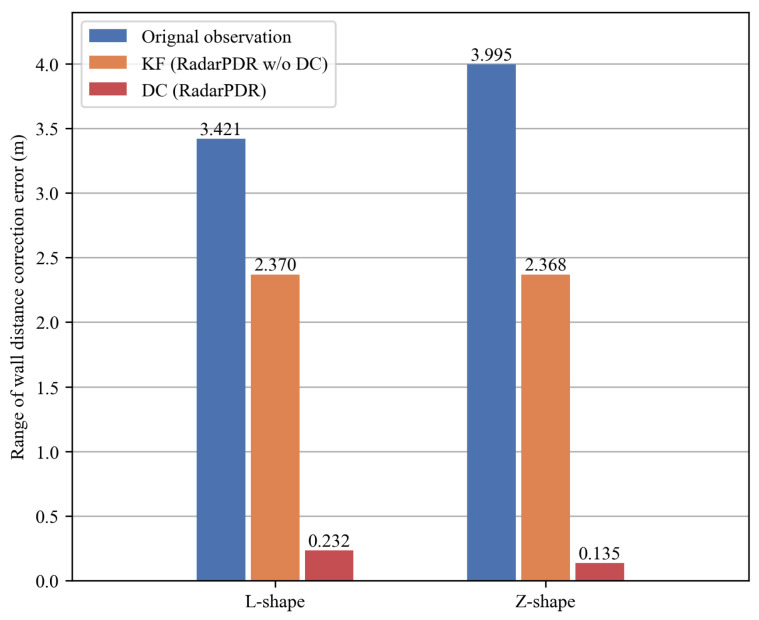
Wall distance error range of all the testing trajectories on the L-shape and the Z-shape routes.

**Figure 11 sensors-23-02782-f011:**
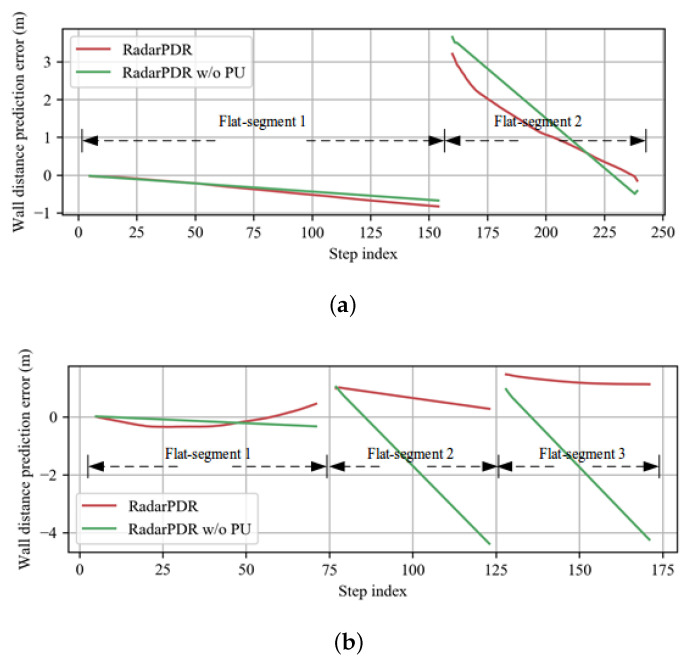
Wall distance prediction error. (**a**) average WDPE of all the testing trajectories on the L-shape route; (**b**) average WDPE of all the testing trajectories on the Z-shape route.

**Figure 12 sensors-23-02782-f012:**
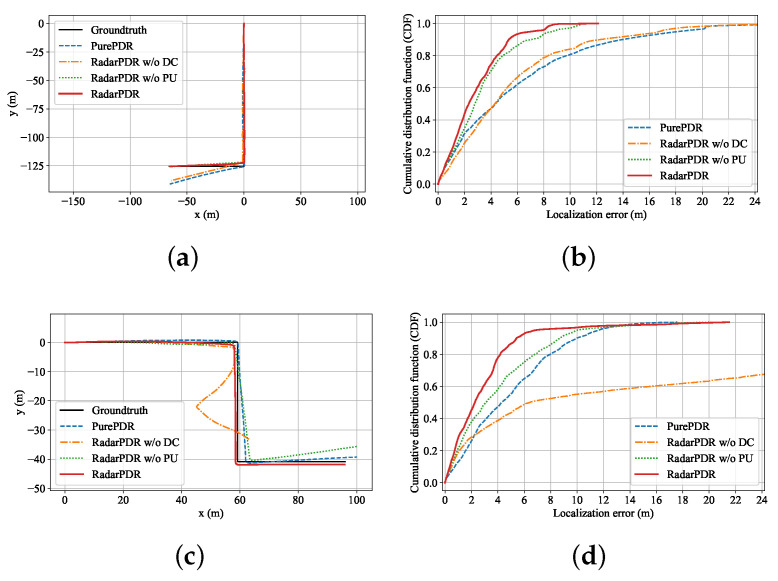
Trajectory and its CDF of ALE. (**a**) average trajectory of all the tests on the L-shape route; (**b**) the localization error CDF of all the testing trajectories on the L-shape route; (**c**) average trajectory of all the tests on the Z-shape route; (**d**) the localization error CDF of all the testing trajectories on the Z-shape route.

## Data Availability

Not applicable.

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
