# Peer review of "RadarPDR: Radar-Assisted Indoor Pedestrian Dead Reckoning"

_sensors, 2023, doi:10.3390/s23052782_

Round 1

Reviewer 1 Report

The paper proposes a method for indoor localization fusing PDR and FMCW radar. The proposed method is validated by experimental results. 

Main comments:

- The novelty of the work should be better framed better. The considered problem is similar to the SLAM problem, which the author do not even mention and discuss in the paper. Further, the authors are not the first to use radard for indoor positioning. Therefore, the other papers that used a radar for indoor positioning (even if not for PDR) from the literature should be cited and compared. 

- The entire text from line 28 to line 40 is a repetition.

- the authors used a stand to support the radar and assumed that the ranging direction is always at 90 degrees with respect to the heading direction. This implies that the authors assume that the person carrying the phone always keeps the phone in the same orientation, equal to the heading. This is not entirely realistic for a practical application. This aspect should be discussed. 

- the authors should describe how the threshold T_v is chosen and the impact of threshold choice on the performance of the alorithm, e.g. the probability of false alarm and probability of detection. 

- In Section 4.3.1. References for the particle filter should be provided at the beginning of the section. 

- throughout the paper, there should be a whitespace between a number and its corresponding unit of measurement, e.g. "10Hz" should be replaced by "10 Hz" and so on.

- line 333: "process noise Q_n" should be replaced by "process noise variance Q_n", and the same for the observation noise.

Author Response

Thank you very much for the valuable comments and kind suggestions on our article. According to these comments, we carefully revise our manuscript to meet the high standards of your journal and provide our point-to-point responses to the comments in the attachment. 

Reviewer 2 Report

The paper is well written but the research is based on quite old reference. the most latest references included is from 2020 and that is also only one. there is no reference from the work done in past 2 years. So the whole paper should be updated in the context of the latest trends and research. Also the comparison must be conducted with the publications from the most recent years to prove the effectiveness of the proposed algorithm.

Author Response

(The authors gave the same response as above.)

Round 2

Reviewer 1 Report

The authors successfully addressed the remarks made in the previous round of review. 

Reviewer 2 Report

The authors have answered all the concerns.